# Exploring DNA Damage and Repair Mechanisms: A Review with Computational Insights

**DOI:** 10.3390/biotech13010003

**Published:** 2024-01-16

**Authors:** Jiawei Chen, Ravi Potlapalli, Heng Quan, Lingtao Chen, Ying Xie, Seyedamin Pouriyeh, Nazmus Sakib, Lichao Liu, Yixin Xie

**Affiliations:** 1College of Letter and Science, University of California, Berkeley, CA 94720, USA; jc01@berkeley.edu; 2College of Computing and Software Engineering, Kennesaw State University, Marietta, GA 30060, USA; lchen25@students.kennesaw.edu (L.C.); rpotlapa@students.kennesaw.edu (R.P.); yxie2@kennesaw.edu (Y.X.); spouriye@kennesaw.edu (S.P.); nsakib1@kennesaw.edu (N.S.); 3Department of Civil and Urban Engineering, New York University, New York, NY 11201, USA; hq322@nyu.edu; 4Stanford Cardiovascular Institute, Stanford University School of Medicine, Palo Alto, CA 94304, USA; lcliu@stanford.edu

**Keywords:** cancer, aging-related diseases, DNA damage, DNA repair mechanisms, uracil-DNA glycosylase, computational biology, DNA database

## Abstract

DNA damage is a critical factor contributing to genetic alterations, directly affecting human health, including developing diseases such as cancer and age-related disorders. DNA repair mechanisms play a pivotal role in safeguarding genetic integrity and preventing the onset of these ailments. Over the past decade, substantial progress and pivotal discoveries have been achieved in DNA damage and repair. This comprehensive review paper consolidates research efforts, focusing on DNA repair mechanisms, computational research methods, and associated databases. Our work is a valuable resource for scientists and researchers engaged in computational DNA research, offering the latest insights into DNA-related proteins, diseases, and cutting-edge methodologies. The review addresses key questions, including the major types of DNA damage, common DNA repair mechanisms, the availability of reliable databases for DNA damage and associated diseases, and the predominant computational research methods for enzymes involved in DNA damage and repair.

## 1. Introduction

DNA damage occurs at a high rate per second, and it causes a change in the genetic information [1,2,3,4]. This may cause cell loss or even the transformation of normal cells to cancer cells. Each cell suffers ten thousand to one million DNA lesions per day [5]. There are two significant resources of DNA damage: exogenous resources, including X-rays, toxins, viruses, bacteria, etc., and endogenous resources, including reactive oxygen species (ROS) [6]. As DNA damage is harmful to normal cells and a significant threat to human health, various mechanisms in DNA repair fix the damage and errors that occur in different cell processes [7,8]. An example of DNA damage is the uracil replacement of cytosine caused by spontaneous deamination, usually excised from DNA by the enzyme uracil-DNA glycosylase. There are several DNA repair pathways to fix DNA damages, including nucleotide excision repair (NER), base excision repair (BER), and mismatch repair (MMR), which are active in different cell cycle stages [9,10]. In addition to those three, homologous recombination (HR) and non-homologous end joining (NHEJ) are discussed frequently in research on DNA damage and repair [11].

The main reason why we care about DNA damage and repair is that DNA damage and sub-optimal DNA damage response (DDR) events lead to diseases, including neurodegenerative diseases that are classified into three types, chromosomal disorders, multifactorial disorders, monogenic disorders: (1) chromosomal disorders, such as Cockayne syndrome; (2) multifactorial disorders, such as Alzheimer’s disease; (3) cancers, such as breast cancer; and (4) monogenic disorders, like ataxia–telangiectasia, age-related macular degeneration, heart disease, etc. [12,13,14]. Endogenous or exogenous cellular processes cause all these diseases mentioned above. The oxidation of nitrogen bases and generation of reactive oxygen species disrupt DNA strands, the alkylation of bases, and hydrolysis, including deamination, depurination, and depyrimidination. The development of bulky adducts is an example of endogenous biological activities, the mismatch of bases because of errors in DNA replication, and monoadduct damage due to a change in the mononitrogen base. DNA adduct damage also results in diseases like diabetes, Parkinson’s disease, heart disease, and atherosclerosis [15,16]. Industrial chemicals such as vinyl chloride, polycyclic aromatic hydrocarbons, and hydrogen peroxide lead to diseases like hereditary diseases, macular degeneration, and sporadic cancer [15]. Research on DNA repair enzymes has been performed and studied since the 1970s [17,18]. The enzymes concerned in DNA restoration are methylguanine methyltransferase, uracil-DNA glycosylase, DNA polymerase β, poly (ADP-ribose) polymerase-1 [6,19], and DNA ligase.

In addition to wet lab research approaches, such as an information management system for clinical genome sequencing [18,19], computational studies are also essential in investigating DNA damage and repair. Computational studies have become effective and efficient with the fast development of computer technologies. They show extraordinary abilities and potential in dealing with large-scale data in DNA study, and they fasten the discovery of biological mechanisms. In computational studies, a database is important in managing and organizing the data, which can provide a structured framework of data to help researchers better understand and access them more efficiently [20]. In DNA repair studies, databases or datasets of human diseases correlate with gene mutations relevant to DNA integrity, stability, and information about DNA damage caused by mutagenic agents [21]. So, in this review article, we also included DNA-related databases and computational methods that have been used for decades in DNA research (Figure 1).

## 2. DNA Repair Mechanisms

The effective response to DNA damage necessitates the coordinated involvement of numerous factors. The integrity of the genome must be preserved, and any potentially harmful mutations that could lead to cellular damage or tumor formation must be prevented. It is crucial to establish a background context that facilitates efficient repair by signaling the presence of DNA lesions. Since DNA repair mechanisms can also be employed as anti-cancer treatments in medical practice, various genotoxic chemicals have been used for several years to cause DNA damage [21]. DNA repair can occur through multiple mechanisms, such as BER, NER, MMR, and DSBR. The categories of DNA damage and repair are shown in Figure 2. 

### 2.1. BER Mechanism

The primary repair process to eliminate DNA damage is the base excision repair pathway, or BER [22,23]. The BER (base excision repair) process is used when DNA is damaged by reactive oxygen species, single-strand breaks, or alkylating agents through oxidation. The key steps in the BER process are as follows: (1) recognition of damage, which utilizes DNA glycosylase enzyme, and each DNA glycosylase is specific to particular types of base damage; (2) removal of damaged base: the DNA glycosylase enzyme cleaves the bond between the damaged base and the sugar phosphate backbone, leaving the apurinic/apyrimidinic (AP) site, which is also known as an abasic site; (3) AP endonuclease: which recognizes the AP site and makes an incision in the DNA strand and creates a single-strand break with a 3′-OH and a 5′-deoxyribosephosphate termini; (4) DNA polymerase: the enzyme fills in the gap by adding the correct base complementary to the undamaged strand; and (5) DNA ligase: as the last step, DNA ligase seals the nick in the DNA backbone, completing the repair process [20,21,24]. The BER pathway is crucial for maintaining the integrity of the genome by fixing common forms of DNA damage. It is a versatile and efficient repair mechanism, addressing a wide range of DNA lesions to ensure the stability and functionality of the genetic material.

### 2.2. NER Mechanism

The NER (nucleotide excision repair) mechanism is employed to repair the damage by creating large adducts and intra-strand crosslinks when UV light and polycyclic aromatic hydrocarbons damage DNA. The nucleotide excision respirasome, a multi-protein complex, performs the NER process in mammals [25,26,27]. The excision of about twenty-eight nucleotide DNA segments furnishing the damaged site is the primary process in eukaryotic NER [27,28]. The two different sub-pathways of global genome repair (GGR) and transcription-coupled repair (TCR) make up NER in mammalian cells [29,30,31]. The XPC-hHR23 complicated is the primary DNA damage key factor in GGR. Another GGR DNA damage binding factor (DDB) [32,33] is a DNA damage sensor. The TFIIHp62 subunit interacts with the damage recognition complex XPC-HR23B in GGR to transport it to the damaged area [34,35]. In terms of liberating damaged DNA, XPB seems to have less helicase activity than XPD. GGR and TCR need this 36 kDa protein [36,37]. RPA, ERCC1, and TFIIH are all known to interact with damaged DNA. RPA and ERCCC investigations have shown that XPA preferentially binds to damaged DNA. The TFIIH helicase subunits’ generated ssDNA intermediate is stabilized by this protein’s ssDNA binding activity [38]. Following DNA synthesis, a twin insertion occurs on the damaged surface because of the sequential recruitment of the XPG and XPF-ERCC1 nuclear. XPG and XPF-ERCC1 are structurally specific nucleases that prefer to hydrolyze double-stranded substrates near ssDNA and sDNA junctions [39], ensuring the proper localization of these proteins to the site of injury and stimulating their junction-breaking endonuclease activity [40]. The DNA substrate bladder, valves, arms, and stem loops are only a few of the DNA substrates that XPG, a 133 kDa protein, affects [41]. The XPG protein possesses two highly conserved nucleic acid motifs spaced apart by a region that aids protein interactions. A protein of 37 kDa in molecular weight, PCNA belongs to the DNA sliding clamp family. In an ATP-dependent process, RFC assembles PCNA on the DNA template by ideally aligning with the 3′-hydroxyl ends of the DNA primer. Polymerase interactions with PCNA and RFC make it possible to synthesize DNA accurately and effectively [42]. POL η and POL ι, in the polymerase β family, exhibit intrinsic exonuclease activity (3′–5′) to correct for reading. The four subunits of mammals form the POL complex are 50, 12, 68, 125, and 68 kDa [43].

### 2.3. MMR Mechanism

The MMR (mismatch excision repair) process is employed when a mismatch develops between bases, such as the T-C and A-G pairs. This is accomplished by removing a strand, which is digested and replaced. DNA mismatch repair (MMR) is the main post-replicative DNA repair mechanism that can increase replication fidelity by 1000 ss [44,45]. Cells exposed to external chemicals and physical agents over time develop DNA damage. Multiple ways exist inside cells to repair DNA damage and stop mutations [44,46]. The DNA resynthesis and MMR initiation processes are thought to involve PCNA [47]. Localizing MutS and MutS to mispairs in freshly duplicated DNA may be made easier by PCNA. Both 5′ and 3′ directed MMR involve the 5′–3′ exonuclease EXO1. High flexibility group box 1 protein (HMGB1 (High Mobility Group Box 1)), RPA, RFC, DNA pol δ, and HMGB1 are other proteins connected to MMR [48]. RPA participates in the entire process of MMR since it attaches to crushed heteroduplex DNA before MutS and MutL, promotes mismatch-provoked excision, guards the ssDNA-gapped region generated after excision, and facilitates DNA synthesis. Additionally, MMR proteins have been associated with homoeologous recombination, immunoglobulin elegance switching, hypermutation, interstrand–crosslink restoration, and trinucleotide repeat (TNR) expansion [49,50]. The MMR employs double-strand DNA breaks produced using uracil DNA glycosylase to restore the AID-triggered G-U mispairs in a strand-indiscriminate manner [51].

## 3. DNA-Related Database

When studying DNA, it is crucial to have databases and datasets to learn the correlation of human diseases with gene mutations relevant to DNA integrity, stability, and information about DNA damage caused by mutagenic agents. Except for some databases that are no longer, here are some valuable databases and datasets in the DNA repair area. Among all the available databases, REPAIRtoire, Reactome, and the KEGG are the most commonly used databases. Table 1 shows the examples of commonly used databases.

### 3.1. REPAIRtoire

REPAIRtoire is a database of repair pathways for protein-coding genes. It provides a comprehensive and curated collection of genetic and epigenetic events that lead to the restoration of normal gene function in human cells. The database includes information on various types of repair mechanisms, such as DNA repair, RNA repair, and protein repair, and it can be used to aid in understanding disease mechanisms and developing new therapeutic strategies [52]. Researchers can search data through the following five sections in this database: (i) proteins: by searching the protein name, you can find the alternative names of the protein, the species of the protein, repair activities, the families of the proteins, and its related diseases; (ii) damage: by searching the name of the DNA damage, you can find the sources of the DNA damage and its effects, and it recognizes proteins; (iii) disease: you can find the related proteins with the disease name; (iv) pathways: the pathways section allows you to access to data through eight pathways from three species (homosapiens, saccharomyces cerevisiae, and escherichia coli); and (v) publications: this section gives you the literature references to entries in the PubMed database. This database lets you quickly search data by entering protein sequences, profile searches, and browsing keywords to find the protein. The links button will give you access to other DNA repair-related databases (REACTOME, KEGG, etc.). REPAIRtoire can be accessed at https://repairtoire.genesilico.pl/ (accessed on 3 January 2023).

### 3.2. Human DNA Repair Genes

Human DNA Repair Genes is a database of a table of human genes. The structure of this database is a table with columns of “Gene Name”, “Activity”, “Chromosome location”, and “Accession number” [53]. This table also has clearly distinguished different sections; it categorized different components and processes involved in DNA repair mechanisms, such as base excision repair (BER) and strand break joining factors, poly (ADP-ribose) polymerase (PARP) enzymes, the section of mismatch excision repair (MMR), chromatin structure, etc. The Human DNA Repair Genes database can be accessed at https://www.mdanderson.org/documents/Labs/Wood-Laboratory/human-dna-repair-genes.html (accessed on 3 January 2023).

### 3.3. Reactome

Like an online laboratory, Reactome is a user-friendly database of huge human pathways and sub-pathways. The main functions of this database are the pathway browser, analysis tools, Reactome FIViz, and documentation. By searching the name, ID, or the location of the gene in the search engine, this database will show you an overview of the topic of DNA repair; after clicking on the event, it will guide you through the pathway browser and show you a mind map, where you can find the description, molecules, structures, and analysis [54]. Here is the link to the DNA repair section: https://reactome.org/content/detail/R-DRE-73894 (accessed on 3 January 2023).

### 3.4. DNArepairK

A database called DNArepairK tracks the kinetics of 70 fluorescently titled DNA repair proteins’ recruitment and clearance from complicated DNA damage sites in vivo in HeLa Kyoto cells. It offers some simple analyses of the dynamics of proteins involved in different DNA repair processes using an interactive graph complemented with live cell imaging movie facilities. Most DNA repair proteins are represented by their kinetics in cells that have not been treated and cells that have been treated with the PARP1/2 inhibitor. This gives an unprecedented overview of how anti-cancer medications affect the regular dynamics of the DNA damage response. Scientists may investigate the DNA damage response using the unique dataset in DNA repair, which will also help them develop and test new anti-cancer medications that target DNA repair [55] DNArepairK can be accessed at http://dnarepair.bas.bg/index.php/dnarepairk/ (accessed on 3 January 2023).

### 3.5. KEGG

The KEGG is a comprehensive database for computer representation of biological systems from cell to organism and the ecosystem; the information is from the genomic to molecular level. This integrated database has been categorized by different information into 16 databases. The molecular networks include the interaction between molecular, reaction, and relation networks featured in the KEGG. The infrastructure of this database is also focused on keeping different organisms’ genes, genomes, and their variations. The database also includes additional types of generalization, such as reaction classes and drug groups [56]. The KEGG can be accessed at https://www.genome.jp/kegg/ (accessed on 3 January 2023).

### 3.6. Brenda

Brenda is a crucial database for primary enzyme functional data collection. The creator of this database extracted data from the primary literature by scientists. The enzymes are categorized by the Enzyme Commission’s list of enzymes. To sort enzyme functional data, you can search enzymes by their EC number, enzyme name, and protein. Some common enzymes with very different properties will share the same EC number, and for more detailed information, you will have to go to the primary literature [57]. This database can be accessed at https://www.brenda-enzymes.info/oldstart.php (accessed on 3 January 2023).

### 3.7. Pathway Commons

Pathway Commons is a public database of public pathway data from several organisms. It contains detailed data on biochemical processes, transport, catalysis activities, complex assembly, and physical interactions involving DNA, complexes, proteins, small molecules, and RNA. This meta-database compiles data from additional databases, like Reactome and Bio Grid. Users have access to a variety of accessible public pathway databases where they may browse and search for routes. Download a comprehensive set of Bio PAX format pathways for a more in-depth study. Additionally, it offers programmers a method to design software for complex investigations [58]. This database can be accessed at http://www.pathwaycommons.org/ (accessed on 3 January 2023).

Like DNA databases, DNA-related databases also have ethical issues. Since the DNA repair databases are publicly accessible, it benefits law enforcement for forensic evidence [59]. Still, the problems of related privacy and human rights may arise at the same time. We should restrict the use of DNA databases for research, investigation, and study purposes, and we need to ask for the data contributor’s acknowledgment to conduct further studies beyond restrictions.

However, most of the above-mentioned databases may contain outdated information. The reasons include that few people are working on building a database, and the data are not easy to collect and analyze. As we noticed in Table 1, the largest amount of data in the KEGG is 45,822,810 because the KEGG is not only a DNA repair-related database but also a vast comprehensive DNA database, as mentioned in Section 3.5. However, some databases (DNArepairK, Human DNA Repair Genes) in Table 1 only contain a few hundred data. As an alternative plan, if a DNA repair database in need is not comprehensive, researchers can refer to other DNA databases; for example, the NCBI GenBank, which is maintained by the National Center for Biotechnology Information, and researchers worldwide use GenBank for a wide range of studies [60].

It is also worth drawing our attention to future database developments. It will be convenient if the database can be frequently updated and provide an easier way to analyze new data. With the help of artificial intelligence, we may have a clever way to automatically collect, analyze data, and manage data in the future.

## 4. DNA Repair Computational Research Methods

Experimental methods tend to be more reliable, time-consuming, and complex to implement. Computational methods offer more cost-effective and efficient ways to explore various scenarios. For example, AlphaFold is a revolutionary advance in the history of protein research that for the first time, offers the practical ability to accurately predict the three-dimensional structure of a protein using amino acid sequences as inputs [61]. While computational methods may not be able to replace laboratory experiments fully, they can help identify and prioritize a selected group of promising candidates from large datasets. Computational studies in biomolecular interactions have proven effective in different topics [59,60,61,62]. This article will introduce three categories of in silico methods based on protein structure, interactions, and evolution.

### 4.1. Protein Structure Analysis

Early efforts to investigate DNA repair enzymes focused on utilizing information from protein structures determined by experimental research. For example, Wang and Moult analyzed missense mutations that cause disease and developed rules based on protein structure stability that could predict the effects of these mutations on molecular function [62]. These rules included the loss of interaction pairs, including hydrogen bonds and salt bridges, the basis of a buried polar residue, the proline insertion into an alpha helix, and the breakage of a disulfide bond. Other researchers also used similar rules [63,64].

Despite significant experimental efforts, the structures of only about 100,000 unique proteins have been resolved, which is a small portion of the known human protein sequences [65]. AlphaFold2 is a tool that has demonstrated success in predicting protein structures. In the 14th Critical Assessment of Protein Structure Prediction (CASP14), a blind test, AlphaFold2, outperformed other prediction methods [66]. While the new prediction algorithm does not fully explain the relationship between a protein’s three-dimensional structure and its sequence, it can accurately predict the structure from the sequence in many cases, making it a practical solution to the protein folding problem [67]. The “SWISS-MODEL” server offers a range of options for constructing homology models, including fully automatic construction through its web interface [68]. In addition to model construction, the server also helps users locate suitable templates and alignments. It is especially useful for modeling proteins highly related to experimentally determined structures, as these relationships can be identified using tools, like “BLAST” [69]. Later, BLAST evolved to BLAST+, and the latest version is “BLAST+ 2.13.0” with advanced features that have made protein structure analysis easier.

### 4.2. Molecular Dynamics Simulations

Molecular dynamics (MD) has become essential in studying DNA repair, allowing for detailed structural and dynamic insights [70]. CHARMM (Chemistry at Harvard Molecular Mechanics), AMBER (Assisted Model Building with Energy Refinement), NAMD (Nanoscale Molecular Dynamics), and GROMACS (Groningen Machine for Chemical Simulations) are well-known software tools that can be used for molecular modeling to study DNA repair and its relationship to proteins and diseases. Many researchers have predicted analyses of protein-DNA and protein–protein contacts by performing molecular dynamics (MD) simulations to support their studies [71,72,73,74], including but not limited to:-Performing molecular dynamics simulations to study the dynamics of proteins involved in DNA repair processes, such as simulating the movement and interactions of DNA repair enzymes;-Studying the effects of genetic variations on the structures and functions of DNA repair proteins, such as simulating the impact of SNPs on the structure and function of DNA repair enzymes;-Predicting the binding process between small molecules and proteins involved in DNA repair to identify potential drug candidates for treating DNA repair-related diseases;-Identifying potential drug candidates for treating and studying the interactions between proteins involved in DNA repair processes, such as simulating the interactions between DNA repair enzymes and DNA damage response proteins.

### 4.3. Evolutionary Analysis

During human population screenings, it was predicted that many amino acid substitution variants found in genes related to DNA repair could reduce protein function and activity [75], potentially leading to reduced repair capacity and an increased risk factor of cancer due to elevated genetic susceptibility. Using evolutionary structure-based methods and sequences makes it possible to differentiate variants depending on a score. Tools such as SIFT, PolyPhen 2.0, and SNAP are commonly used. SIFT predicts the effect of variants as neutral or deleterious using a normalized probability score based on sequence homology. It assumes that important amino acids will be reserved in the protein family, so various changes at well-reserved positions are often predicted to be harmful [76]. PolyPhen 2.0 combines sequence and structure-based attributes and utilizes a naive Bayesian classifier to identify the effect of an amino acid substitution [77]. SNAP is a neural network-based tool that accurately predicts the functional effects of nonsynonymous single nucleotide polymorphisms (nsSNPs) by combining evolutionary information (such as residue conservation within sequence families), predicted aspects of protein structure (such as secondary structure and solvent accessibility), and other pertinent data [78]. These algorithms correctly identified approximately 80% of amino acid substitutions that were assumed to significantly decrease the activity of the variant protein, as tested in the set [78,79].

For ethical considerations, as researchers, we should maintain the integrity and accuracy of data and show respect for the dignity, rights, and privacy of research participants. To avoid risks like privacy leaks, shortly research, we will mainly focus on using a protein data bank, open-source data, for training purposes. We will also implement robust methods to avoid such risks.

Computational methods have their own limitations. They may simplify the representation of biological systems [80]. For example, computational resources often limit molecular dynamics (MD) simulations, restricting the timescales and length scales that can be realistically simulated [81]. Many biological processes, such as protein folding or large-scale conformational changes, occur on timescales beyond the reach of current MD simulations [82,83,84,85,86]. In the protein structural prediction area, the accuracy of computational models heavily relies on the quality and quantity of the data used for training [86,87,88]. Biological systems are dynamic, with molecules constantly moving and interacting. Computational models, including AlphaFold, may not accurately capture these temporal dynamics. Computational models are constrained by existing knowledge and databases. Therefore, if a biological mechanism is poorly understood or novel, the model may struggle to predict the structures or functions associated with that mechanism accurately. Overfitting occurs when a model becomes too specific to the training data, performing poorly on new, unseen data, especially when the training data are limited.

Here are some future developments that may be helpful. We can enhance models to capture the temporal dynamics of biological systems better. Also, future models should be more adaptable to unknown or poorly understood biological mechanisms. We also need to make efforts to enhance data quality and quantity. The future models should be more resource-efficient and enhance generalization across diverse biological systems.

## 5. Summary

DNA damage and repair mechanisms have been studied for decades and will still be considered an important topic in future research related to diseases, aging, cancers, etc. The mutations in DNA repair genes can influence and regulate individual cancer susceptibility, and polymorphism screening has recently become a research area in molecular epidemiology with high potential. Targeted gene therapy is one of the possible methods since it can selectively repair drug sensitivity in cancer cells with drug sensitivity abnormalities. Understanding these mechanisms will aid in the development of new therapeutic approaches for patients with defective tumors, as well as in the choice of treatment and prognosis for ovarian and breast cancer patients. Several forms of cancer are affected by both intrinsic and acquired resistance mechanisms, which is a significant area for the development of new medications. In our review article, we summarized DNA repair enzymes that are responsible for fixing DNA damage, DNA repair databases that are helpful for researchers to study DNA damage and repair mechanisms using data science and data analysis techniques, and DNA repair computational research methods that are preferable for scientists to perform research in different ways. By conducting research in environment chemical research, toxicological genotoxic drugs can lead to a reduction in DNA damage. The numerous clinical trials that evaluate the possibility of making tumors more responsive to chemotherapy by inhibiting RSR signaling may provide directions for future efforts to create tractable drugs that specifically target tumor cells.

## Figures and Tables

**Figure 1 biotech-13-00003-f001:**
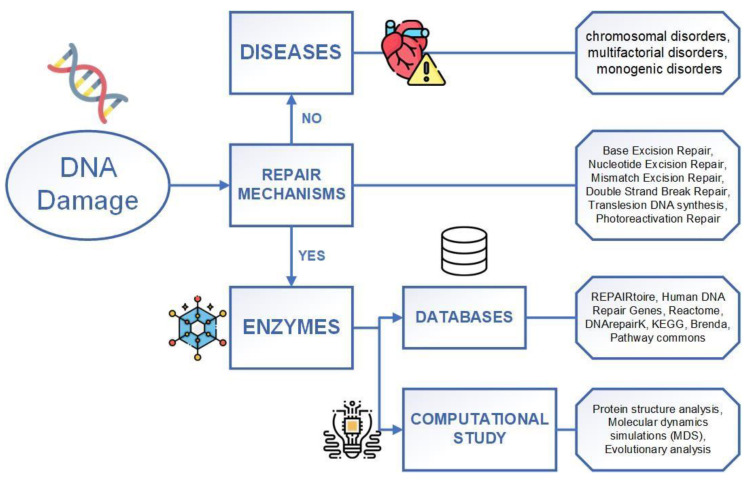
A structural representation of this review paper. It describes how repair mechanisms can repair DNA damage by utilizing enzymes.

**Figure 2 biotech-13-00003-f002:**
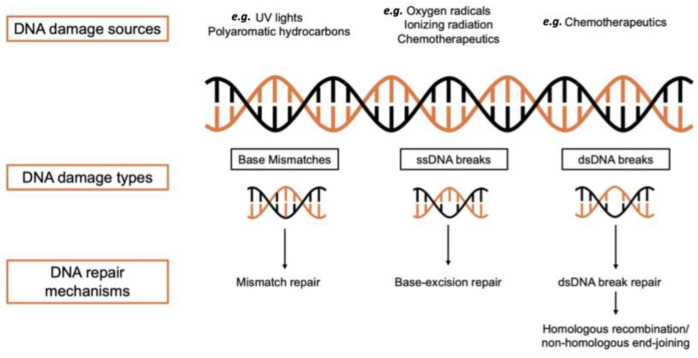
DNA damage types and repair mechanisms. Three major DNA repair mechanisms with their key factors and associated damage types are shown.

**Table 1 biotech-13-00003-t001:** Examples of DNA repair-related databases.

Database	Size	Feature	Function	Last Update
REPAIRtoire	Proteins and DNA damage Diseases: 429	Multi-organism support; gene search	Identify and analyze pathways involved in DNA repair	October 2010
Human DNA Repair Genes	Genes: 256	Categorized DNA repair datasets	Related gene activity and chromosome location	June 2020
Reactome	Curated human protein: 11,350	Visualize biological processes online	Visualization, interpretation, and analysis	November 2022
DNArepairK	Proteins: 72	Animation of DNA repair protein kinetics	Dynamics of DNA repair proteins at the sites of DNA lesions	Unknown
KEGG	45, 822, 810	Large-scale integrated database	Molecular networks and network variants	November 2020
Brenda	Enzymes: 8423	Collection of gene data and enzymes	Searching enzymes	January 2023
Pathway commons	5772 Pathways; 2.3 million interaction data	Multiple databases to collect data	Data downloads, BioPAX web services, and data visualization	January 2020

## Data Availability

REPAIRtoire can be accessed at https://repairtoire.genesilico.pl/ (accessed on 3 January 2023). The Human DNA Repair Genes can be accessed at https://www.mdanderson.org/documents/Labs/Wood-Laboratory/human-dna-repair-genes.html (accessed on 3 January 2023). DNArepairK can be accessed at http://dnarepair.bas.bg/index.php/dnarepairk/ (accessed on 3 January 2023). KEGG can be accessed at https://www.genome.jp/kegg/ (accessed on 3 January 2023). The Brenda database can be accessed at https://www.brenda-enzymes.info/oldstart.php (accessed on 3 January 2023). The pathway database can be accessed at http://www.pathwaycommons.org/ (accessed on 3 January 2023).

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
