# Peer review of "Exploring DNA Damage and Repair Mechanisms: A Review with Computational Insights"

_biotech, 2024, doi:10.3390/biotech13010003_

Round 1

Reviewer 1 Report

Comments and Suggestions for Authors

Dear authors,

The major questions related to DNA-related databases and computational methods that you should answer in this  research may include:

1. What are the most commonly used DNA-related databases in DNA research?
2. How have computational methods contributed to advancements in DNA research?
3. What are the limitations or challenges associated with DNA-related databases and computational methods?
4. What are the ethical considerations surrounding the use of DNA-related databases and computational methods?
5. What are the potential future developments or trends in DNA research databases and computational methods?

Best regards!

Author Response

Thanks for all the comments. We have addressed all the questions and uploaded the manuscript revision. Here are our responses to the reviewer’s questions.

  1. What are the most commonly used DNA-related databases in DNA research?

Thanks for your question. The related contents were updated in Section 3. “DNA-related database,” as highlighted (lines 156-160).

  1. How have computational methods contributed to advancements in DNA research?

Thanks for your question. We have added the contents to describe the significance of computational methods in Section 4. “DNA Repair Computational Research Methods.” These changes were highlighted (lines 262-265).

  1. What are the limitations or challenges associated with DNA-related databases and computational methods?

Thanks for your question. We have added the contents to describe the limitations associated with databases and computational methods in DNA study. These changes have been added in Section 3 and Section 4, as highlighted (lines 244-253, lines 339-352)

  1. What are the ethical considerations surrounding the use of DNA-related databases and computational methods?

Thanks for your question. We have added the discussion at the end of Section 3 and Section 4 as highlighted (lines 237-242, lines 333-337)

  1. What are the potential future developments or trends in DNA research databases and computational methods?

Thanks for your question. We have added the contents to describe the potential future developments of DNA research databases and computational methods in DNA study. These changes have been added in Section 3 and Section 4, as highlighted (lines 255-258, lines 354-358)

Reviewer 2 Report

Comments and Suggestions for Authors

A review by Chen et al. focuses on DNA damage that is a critical factor contributing to genetic alterations, directly affecting human health. As over the past decade, substantial progress and pivotal discoveries have been achieved in DNA damage and repair, this review consolidates research efforts, focusing on DNA repair mechanisms, computational research methods, and associated databases. This work is a resource for scientists and researchers engaged in computational DNA research, offering the latest insights into DNA-related proteins, diseases, and cutting-edge methodologies. 

The majority of references have been published long time ago. The Authors should also refer to more current literature.

Figure 2 - please remove "...". I suppose these mean other undefined factors, however, it is unnecessary and this piece of information can be put e.g., in figure legend that there are more, not yet defined factors.

Table 1 should be formatted for clarity.

Author Response

Thanks for all the comments. We have addressed all the questions and uploaded the manuscript revision. Here are our responses to the reviewer’s questions.

  1. The majority of references have been published long time ago. The Authors should also refer to more current literature.

Thanks for your comments. We have added more literature review content on the latest references on computational methods. They are included in Section 4 as highlighted (e.g., lines 339-352).

  1. Figure 2 - please remove "...". I suppose these mean other undefined factors, however, it is unnecessary and this piece of information can be put e.g., in figure legend that there are more, not yet defined factors.

Thanks for your comments. We have removed “…” in Figure 2; instead, we used “e.g.,” to demonstrate more examples of DNA damage resources.

  1. Table 1 should be formatted for clarity.

Thanks for your comment. We formatted the Table 1 using BioTech requirements on Table.

Round 2

Reviewer 1 Report

Comments and Suggestions for Authors

No other comments